# The Causal-Neural Connection:
# Expressiveness, Learnability, and Inference

**Kevin Xia**
CausalAI Lab
Columbia University
kmx2000@columbia.edu

**Kai-Zhan Lee**
Bloomberg L.P.
Columbia University
kl2792@columbia.edu

**Yoshua Bengio**
MILA
Université de Montréal
yoshua.bengio@mila.quebec

**Elias Bareinboim**
CausalAI Lab
Columbia University
eb@cs.columbia.edu

## Abstract

One of the central elements of any causal inference is an object called structural causal model (SCM), which represents a collection of mechanisms and exogenous sources of random variation of the system under investigation (Pearl, 2000). An important property of many kinds of neural networks is *universal approximability*: the ability to approximate any function to arbitrary precision. Given this property, one may be tempted to surmise that a collection of neural nets is capable of learning any SCM by training on data generated by that SCM. In this paper, we show this is not the case by disentangling the notions of expressivity and learnability. Specifically, we show that the causal hierarchy theorem (Thm. 1, Bareinboim et al., 2020), which describes the limits of what can be learned from data, still holds for neural models. For instance, an arbitrarily complex and expressive neural net is unable to predict the effects of interventions given observational data alone. Given this result, we introduce a special type of SCM called a neural causal model (NCM), and formalize a new type of inductive bias to encode structural constraints necessary for performing causal inferences. Building on this new class of models, we focus on solving two canonical tasks found in the literature known as causal identification and estimation. Leveraging the neural toolbox, we develop an algorithm that is both sufficient and necessary to determine whether a causal effect can be learned from data (i.e., causal identifiability); it then estimates the effect whenever identifiability holds (causal estimation). Simulations corroborate the proposed approach.

## 1 Introduction

One of the most celebrated and relied upon results in the science of intelligence is the universality of neural models. More formally, universality says that neural models can approximate any function (e.g., boolean, classification boundaries, continuous valued) with arbitrary precision given enough capacity in terms of the depth and breadth of the network [14, 26, 47, 53]. This result, combined with the observation that most tasks can be abstracted away and modeled as input/output – i.e., as functions – leads to the strongly held belief that under the right conditions, neural networks can solve the most challenging and interesting tasks in AI. This belief is not without merits, and is corroborated by ample evidence of practical successes, including in compelling tasks in computer vision [43], speech recognition [22], and game playing [54]. Given that the universality of neural nets is such a compelling proposition, we investigate this belief in the context of causal reasoning.

To start understanding the causal-neural connection – i.e., the non-trivial and somewhat intricate relationship between these modes of reasoning – two standard objects in causal analysis will be instrumental. First, we evoke a class of generative models known as the *Structural Causal Model* (SCM, for short) [58, Ch. 7]. In words, an SCM $\mathcal{M}^*$ is a representation of a system that includes a collection of mechanisms and a probability distribution over the exogenous conditions (to be formally defined later on). Second, any fully specified SCM $\mathcal{M}^*$ induces a collection of distributions known as the *Pearl Causal Hierarchy* (PCH) [5, Def. 9]. The importance of the PCH is that it formally delimits

35th Conference on Neural Information Processing Systems (NeurIPS 2021).

distinct cognitive capabilities (also known as layers; not to be confused with neural nets layers) that can be associated with the human activities of "seeing" (layer 1), "doing" (2), and "imagining" (3) [59, Ch. 1]. [1] Each of these layers can be expressed as a distinct formal language and represents queries that can help to classify different types of inferences [5, Def. 8]. Together, these layers form a strict containment hierarchy [5, Thm. 1]. We illustrate these notions in Fig. 1(a) (left side), where SCM $\mathcal{M}^*$ induces layers $L_1^*, L_2^*, L_3^*$ of the PCH.

Even though each possible statement within these capabilities has well-defined semantics given the true SCM $\mathcal{M}^*$ [58, Ch. 7], a challenging inferential task arises when one wishes to recover part of the PCH when $\mathcal{M}^*$ is only partially observed. This situation is typical in the real world aside from some special settings in physics and chemistry where the laws of nature are understood with high precision.

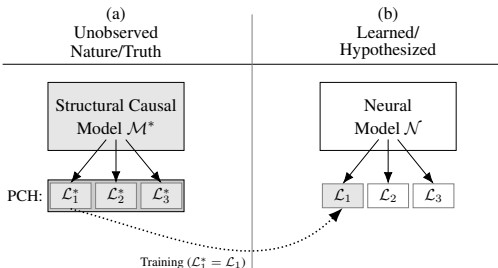

For concreteness, consider the setting where one needs to make a statement about the effect of a new intervention (i.e., about layer 2), but only has observational data from layer 1, which is passively collected.[2] Going back to the causal-neural connection, one could try to learn a neural model $\mathcal{N}$ using the observational dataset (layer 1) generated by the true SCM $\mathcal{M}^*$, as illustrated in Fig. 1(b). Naturally, a basic consistency requirement is that $\mathcal{N}$ should be capable of generating the same distributions as $\mathcal{M}^*$; in this case, their layer 1 predictions should match (i.e., $L_1 = L_1^*$). Given the universality of neural models, it is not hard to believe that these constraints can be satisfied in the large sample limit. The question arises of whether the learned model $\mathcal{N}$ can act as a proxy, having the capability of predicting the effect of interventions that matches the $L_2$ distribution generated by the true (unobserved) SCM $\mathcal{M}^*$. [3] The answer to this question cannot be ascertained in general, as will become evident later on (Corol. 1). The intuitive reason behind this result is that there are multiple neural models that are equally consistent w.r.t. the $L_1$ distribution of $\mathcal{M}^*$ but generate different $L_2$-distributions. [4] Even though $\mathcal{N}$ may be expressive enough to fully represent $\mathcal{M}^*$ (as discussed later on), generating one particular parametrization of $\mathcal{N}$ consistent with $L_1$ is insufficient to provide any guarantee regarding higher-layer inferences, i.e., about predicting the effects of interventions ($L_2$) or counterfactuals ($L_3$).

The discussion above entails two tasks that have been acknowledged in the literature, namely, causal effect identification and estimation. The first – causal identification – has been extensively studied, and general solutions have been developed, such as Pearl's celebrated do-calculus [57]. Given the impossibility described above, the ingredient shared across current non-neural solutions is to represent assumptions about the unknown $\mathcal{M}^*$ in the form of causal diagrams [58, 65, 7] or their equivalence classes [28, 60, 29, 71]. The task is then to decide whether there is a unique solution for the causal query based on such assumptions. There are no neural methods today focused on solving this task.

The second task – causal estimation – is triggered when effects are determined to be identifiable by the first task. Whenever identifiability is obtained through the backdoor criterion/conditional ignorability [58, Sec. 3.3.1], deep learning techniques can be leveraged to estimate such effects with impressive practical performance [63, 52, 48, 31, 69, 70, 35, 64, 15, 25, 37, 30]. For effects that are identifiable through causal functionals that are not necessarily of the backdoor-form (e.g., frontdoor,

Figure 1: The l.h.s. contains the unobserved true SCM $\mathcal{M}^*$ that induces the three layers of the PCH. The r.h.s. contains an NCM that is trained to match in layer 1. The matching shading indicates that the two models agree w.r.t. $L_1$ while not necessarily agreeing w.r.t. layers 2 and 3.

---

[1]This structure is named after Judea Pearl and is a central topic in his *Book of Why (BoW)*, where it is also called the "Ladder of Causation" [59]. For a more technical discussion on the PCH, we refer readers to [5].

[2]The full inferential challenge is, in practice, more general since an agent may be able to perform interventions and obtain samples from a subset of the PCH's layers, while its goal is to make inferences about some other parts of the layers [7, 46, 5]. This situation is not uncommon in RL settings [66, 17, 44, 45]. Still, for the sake of space and concreteness, we will focus on two canonical and more basic tasks found in the literature.

[3]We defer a more formal discussion on how neural models could be used to assess the effect of interventions to Sec. 2. Still, this is neither attainable in all universal neural architectures nor trivially implementable.

[4]Pearl shared a similar observation in the *BoW* [59, p. 32]: "Without the causal model, we could not go from rung (layer) one to rung (layer) two. This is why deep-learning systems (as long as they use only rung-one data and do not have a causal model) will never be able to answer questions about interventions (...)".

napkin), other optimization/statistical techniques can be employed that enjoy properties such as double robustness and debiasedness [32, 33, 34]. Each of these approaches optimizes a particular estimand corresponding to one specific target interventional distribution.

Despite all the great progress achieved so far, it is still largely unknown how to perform the tasks of causal identification and estimation in arbitrary settings using neural networks as a generative model, acting as a proxy for the true SCM $\mathcal{M}^*$. It is our goal here to develop a general causal-neural framework that has the potential to scale to real-world, high-dimensional domains while preserving the validity of its inferences, as in traditional symbolic approaches. In the same way that the causal diagram encodes the assumptions necessary for the do-calculus to decide whether a certain query is identifiable, our method encodes the same invariances as an inductive bias while being amenable to gradient-based optimization, allowing us to perform both tasks in an integrated fashion (in a way, addressing Pearl's concerns alluded to in Footnote 4). Specifically, our contributions are as follows:

1. [Sec. 2] We introduce a special yet simple type of SCM that is amenable to gradient descent called a *neural causal model* (NCM). We prove basic properties of this class of models, including its universal expressiveness and ability to encode an inductive bias representing certain structural invariances (Thm. 1-3). Notably, we show that despite the NCM's expressivity, it still abides by the Causal Hierarchy Theorem (Corol. 1).

2. [Sec. 3] We formalize the problem of neural identification (Def. 8) and prove a duality between identification in causal diagrams and in neural causal models (Thm. 4). We introduce an operational way to perform inferences in NCMs (Corol. 2-3) and a sound and complete algorithm to jointly train and decide effect identifiability for an NCM (Alg. 1, Corol. 4).

3. [Sec. 4] Building on these results, we develop a gradient descent algorithm to jointly identify and estimate causal effects (Alg. 2).

There are multiple ways of grounding these theoretical results. In Sec. 5, we perform experiments with one possible implementation which support the feasibility of the proposed approach. All appendices including proofs, experimental details, and examples can be found in the full technical report [68].

## 1.1 Preliminaries

In this section, we provide the necessary background to understand this work, following the presentation in [58]. An uppercase letter $X$ indicates a random variable, and a lowercase letter $x$ indicates its corresponding value; bold uppercase $\mathbf{X}$ denotes a set of random variables, and lowercase letter $\mathbf{x}$ its corresponding values. We use $\mathcal{D}_X$ to denote the domain of $X$ and $\mathcal{D}_\mathbf{X} = \mathcal{D}_{X_1} \times \cdots \times \mathcal{D}_{X_k}$ for $\mathbf{X} = \{X_1, \ldots, X_k\}$. We denote $P(\mathbf{X})$ as a probability distribution over a set of random variables $\mathbf{X}$ and $P(\mathbf{X} = \mathbf{x})$ as the probability of $\mathbf{X}$ being equal to the value of $\mathbf{x}$ under the distribution $P(\mathbf{X})$. For simplicity, we will mostly abbreviate $P(\mathbf{X} = \mathbf{x})$ as simply $P(\mathbf{x})$. The basic semantic framework of our analysis rests on *structural causal models* (SCMs) [58, Ch. 7], which are defined below.

**Definition 1** (Structural Causal Model (SCM)). An SCM $\mathcal{M}$ is a 4-tuple $\langle \mathbf{U}, \mathbf{V}, \mathcal{F}, P(\mathbf{U}) \rangle$, where $\mathbf{U}$ is a set of exogenous variables (or "latents") that are determined by factors outside the model; $\mathbf{V}$ is a set $\{V_1, V_2, \ldots, V_n\}$ of (endogenous) variables of interest that are determined by other variables in the model – that is, in $\mathbf{U} \cup \mathbf{V}$; $\mathcal{F}$ is a set of functions $\{f_{V_1}, f_{V_2}, \ldots, f_{V_n}\}$ such that each $f_i$ is a mapping from (the respective domains of) $\mathbf{U}_{V_i} \cup \mathbf{Pa}_{V_i}$ to $V_i$, where $\mathbf{U}_{V_i} \subseteq \mathbf{U}$, $\mathbf{Pa}_{V_i} \subseteq \mathbf{V} \setminus V_i$, and the entire set $\mathcal{F}$ forms a mapping from $\mathbf{U}$ to $\mathbf{V}$. That is, for $i = 1, \ldots, n$, each $f_i \in \mathcal{F}$ is such that $v_i \leftarrow f_{V_i}(\mathbf{pa}_{V_i}, \mathbf{u}_{V_i})$; and $P(\mathbf{u})$ is a probability function defined over the domain of $\mathbf{U}$. ∎

Each SCM $\mathcal{M}$ induces a causal diagram $G$ where every $V_i \in \mathbf{V}$ is a vertex, there is a directed arrow $(V_j \to V_i)$ for every $V_i \in \mathbf{V}$ and $V_j \in Pa(V_i)$, and there is a dashed-bidirected arrow $(V_j \dashleftarrow\dashrightarrow V_i)$ for every pair $V_i, V_j \in \mathbf{V}$ such that $\mathbf{U}_{V_i}$ and $\mathbf{U}_{V_j}$ are not independent. For further details on this construction, see [5, Def. 13/16, Thm. 4]. The exogenous $\mathbf{U}_{V_i}$'s are not assumed independent (i.e. Markovianity does not hold). We will consider here *recursive* SCMs, which implies acyclic diagrams, and that the endogenous variables ($\mathbf{V}$) are discrete and have finite domains.

We show next how an SCM $\mathcal{M}$ gives values to the PCH's layers; for details on the semantics, see [5, Sec. 1.2]. Superscripts are omitted when unambiguous.

**Definition 2** (Layers 1, 2 Valuations). An SCM $\mathcal{M}$ induces layer $L_2(\mathcal{M})$, a set of distributions over $\mathbf{V}$, one for each intervention $\mathbf{x}$. For each $\mathbf{Y} \subseteq \mathbf{V}$,

$$P^{\mathcal{M}}(\mathbf{y}_\mathbf{x}) = \sum_{\{\mathbf{u} | \mathbf{Y}_\mathbf{x}(\mathbf{u}) = \mathbf{y}\}} P(\mathbf{u}), \tag{1}$$

where $\mathbf{Y_x(u)}$ is the solution for $\mathbf{Y}$ after evaluating $\mathcal{F_x} := \{f_{V_i} : V_i \in \mathbf{V} \setminus \mathbf{X}\} \cup \{f_X \leftarrow x : X \in \mathbf{X}\}$. The specific distribution $P(\mathbf{V})$, where $\mathbf{X}$ is empty, is defined as layer $L_1(\mathcal{M})$. ∎

In words, an external intervention forcing a set of variables $\mathbf{X}$ to take values $\mathbf{x}$ is modeled by replacing the original mechanism $f_X$ for each $X \in \mathbf{X}$ with its corresponding value in $\mathbf{x}$. This operation is represented formally by the do-operator, $do(\mathbf{X} = \mathbf{x})$, and graphically as the *mutilation* procedure. For the definition of the third layer, $L_3(\mathcal{M})$, see Def. 9 in Appendix A or [5, Def. 7].

## 2  Neural Causal Models and the Causal Hierarchy Theorem

In this section, we aim to resolve the tension between expressiveness and learnability (Fig. 1). To that end, we define a special class of SCMs based on neural nets that is amenable to optimization and has the potential to act as a proxy for the true, unobserved SCM $\mathcal{M}^*$.

**Definition 3** (NCM). A Neural Causal Model (for short, NCM) $\widehat{M}(\boldsymbol{\theta})$ over variables $\mathbf{V}$ with parameters $\boldsymbol{\theta} = \{\theta_{V_i} : V_i \in \mathbf{V}\}$ is an SCM $\langle \widehat{\mathbf{U}}, \mathbf{V}, \widehat{\mathcal{F}}, P(\widehat{\mathbf{U}}) \rangle$ such that

- $\widehat{\mathbf{U}} \subseteq \{\widehat{U}_{\mathbf{C}} : \mathbf{C} \subseteq \mathbf{V}\}$, where each $\widehat{U}$ is associated with some subset of variables $\mathbf{C} \subseteq \mathbf{V}$, and $\mathcal{D}_{\widehat{U}} = [0, 1]$ for all $\widehat{U} \in \widehat{\mathbf{U}}$. (Unobserved confounding is present whenever $|\mathbf{C}| > 1$.)
- $\widehat{\mathcal{F}} = \{\hat{f}_{V_i} : V_i \in \mathbf{V}\}$, where each $\hat{f}_{V_i}$ is a feedforward neural network parameterized by $\theta_{V_i} \in \boldsymbol{\theta}$ mapping values of $\mathbf{U}_{V_i} \cup \mathbf{Pa}_{V_i}$ to values of $V_i$ for some $\mathbf{Pa}_{V_i} \subseteq \mathbf{V}$ and $\mathbf{U}_{V_i} = \{\widehat{U}_{\mathbf{C}} : \widehat{U}_{\mathbf{C}} \in \widehat{\mathbf{U}}, V_i \in \mathbf{C}\}$;
- $P(\widehat{\mathbf{U}})$ is defined s.t. $\widehat{U} \sim \mathrm{Unif}(0, 1)$ for each $\widehat{U} \in \widehat{\mathbf{U}}$. ∎

There is a number of remarks worth making at this point.

1. **[Relationship NCM → SCM]** By definition, all NCMs are SCMs, which means NCMs have the capability of generating any distribution associated with the PCH's layers.

2. **[Relationship SCM ↛ NCM]** On the other hand, not all SCMs are NCMs, since Def. 3 dictates that $\widehat{\mathbf{U}}$ follows uniform distributions in the unit interval and $\widehat{\mathcal{F}}$ are feedforward neural networks.

3. **[Non-Markovianity]** For any two endogenous variables $V_i$ and $V_j$, it is the case that $\mathbf{U}_{V_i}$ and $\mathbf{U}_{V_j}$ might share an input from $\widehat{\mathbf{U}}$, which will play a critical role in causality, not ruling out *a priori* the possibility of unobserved confounding and violations of Markovianity.

4. **[Universality of Feedforward Nets]** Feedforward networks are universal approximators [14, 26] (see also [19]), and any probability distribution can be generated by the uniform one (e.g., see *probability integral transform* [1]). This suggests that the pair $\langle \widehat{\mathcal{F}}, P(\widehat{\mathbf{U}}) \rangle$ may be expressive enough for modeling $\mathcal{M}^*$'s mechanisms $\mathcal{F}$ and distribution $P(\mathbf{U})$ without loss of generality.

5. **[Generalizations / Other Model Classes]** The particular modeling choices within the definition above were made for the sake of explanation, and the results discussed here still hold for other, arbitrary classes of functions and probability distributions, as shown in Appendix D.

To compare the expressiveness of NCMs and SCMs, we introduce the following definition.

**Definition 4** ($\mathrm{P}^{(L_i)}$-Consistency). Consider two SCMs, $\mathcal{M}_1$ and $\mathcal{M}_2$. $\mathcal{M}_2$ is said to be $\mathrm{P}^{(L_i)}$-consistent (for short, $L_i$-consistent) w.r.t. $\mathcal{M}_1$ if $L_i(\mathcal{M}_1) = L_i(\mathcal{M}_2)$. ∎

This definition applies to NCMs since they are also SCMs. As shown below, NCMs can not only approximate the collection of functions of the true SCM $\mathcal{M}^*$, but they can perfectly *represent* all the observational, interventional, and counterfactual distributions. This property is, in fact, special and not enjoyed by many neural models. (For examples and discussion, see Appendix C and D.1.)

**Theorem 1** (NCM Expressiveness). *For any SCM $\mathcal{M}^* = \langle \mathbf{U}, \mathbf{V}, \mathcal{F}, P(\mathbf{U}) \rangle$, there exists an NCM $\widehat{M}(\boldsymbol{\theta}) = \langle \widehat{\mathbf{U}}, \mathbf{V}, \widehat{\mathcal{F}}, P(\widehat{\mathbf{U}}) \rangle$ s.t. $\widehat{M}$ is $L_3$-consistent w.r.t. $\mathcal{M}^*$.* ∎

Thm. 1 ascertains that there is no loss of expressive power using NCMs despite the constraints imposed over its form, i.e., NCMs are as expressive as SCMs. One might be tempted to surmise, therefore, that an NCM can be trained on the observed data and act as a proxy for the true SCM $\mathcal{M}^*$, and inferences about other quantities of $\mathcal{M}^*$ can be done through computation directly in $\widehat{\mathcal{M}}$. Unfortunately, this is almost never the case: [5]

---

[5]Multiple examples of this phenomenon are discussed in Appendix C.1 and [5, Sec. 1.2]

**Corollary 1** (Neural Causal Hierarchy Theorem (N-CHT)). *Let $\Omega^*$ and $\Omega$ be the sets of all SCMs and NCMs, respectively. We say that Layer $j$ of the causal hierarchy for NCMs collapses to Layer $i$ $(i < j)$ relative to $\mathcal{M}^* \in \Omega^*$ if $L_i(\mathcal{M}^*) = L_i(\widehat{M})$ implies that $L_j(\mathcal{M}^*) = L_j(\widehat{M})$ for all $\widehat{M} \in \Omega$. Then, with respect to the Lebesgue measure over (a suitable encoding of $L_3$-equivalence classes of) SCMs, the subset in which Layer $j$ of NCMs collapses to Layer $i$ has measure zero.* ∎

This corollary highlights the fundamental challenge of performing inferences across the PCH layers even when the target object (NCM $\widehat{\mathcal{M}}$) is a suitable surrogate for the underlying SCM $\mathcal{M}^*$, in terms of expressiveness and capability of generating the same observed distribution. That is, expressiveness does not mean that the learned object has the same empirical content as the generating model. For concrete examples of the expressiveness of NCMs and why it is insufficient for causal inference, see Examples 1 and 2 in Appendix C.1. Thus, structural assumptions are necessary to perform causal inferences when using NCMs, despite their expressiveness. We discuss next how to incorporate the necessary assumptions into an NCM to circumvent the limitation highlighted by Corol. 1.

## 2.1 A Family of Neural-Interventional Constraints (Inductive Bias)

In this section, we investigate constraints about $\mathcal{M}^*$ that will narrow down the hypothesis space and possibly allow for valid cross-layer inferences. One well-studied family of structural constraints comes in the form of a pair comprised of a collection of interventional distributions $\mathcal{P}$ and causal diagram $\mathcal{G}$, known as a *causal bayesian network* (CBN) (Def. 15; see also [5, Thm. 4])). The diagram $\mathcal{G}$ encodes constraints over the space of interventional distributions $\mathcal{P}$ which are useful to perform cross-layer inferences (for details, see Appendix C.2). For simplicity, we focus on interventional inferences from observational data. To compare the constraints entailed by distinct SCMs, we define the following notion of consistency:

**Definition 5** ($\mathcal{G}$-Consistency). Let $\mathcal{G}$ be the causal diagram induced by SCM $\mathcal{M}^*$. For any SCM $\mathcal{M}$, we say that $\mathcal{M}$ is $\mathcal{G}$-consistent (w.r.t. $\mathcal{M}^*$) if $\mathcal{G}$ is a CBN for $L_2(\mathcal{M})$. ∎

In the context of NCMs, this means that $\mathcal{M}$ would impose the same constraints over $\mathcal{P}$ as the true SCM $\mathcal{M}^*$ (since $\mathcal{G}$ is also a CBN for $L_2(\mathcal{M}^*)$ by [5, Thm. 4]). Whenever the corresponding diagram $\mathcal{G}$ is known, one should only consider NCMs that are $\mathcal{G}$-consistent. [6] We provide below a systematic way of constructing $\mathcal{G}$-consistent NCMs.

**Definition 6** ($C^2$-Component). For a causal diagram $\mathcal{G}$, a subset $\mathbf{C} \subseteq \mathbf{V}$ is a complete confounded component (for short, $C^2$-component) if any pair $V_i, V_j \in \mathbf{C}$ is connected with a bidirected arrow in $\mathcal{G}$ and is maximal (i.e. there is no $C^2$-component $\mathbf{C}'$ for which $\mathbf{C} \subset \mathbf{C}'$.) ∎

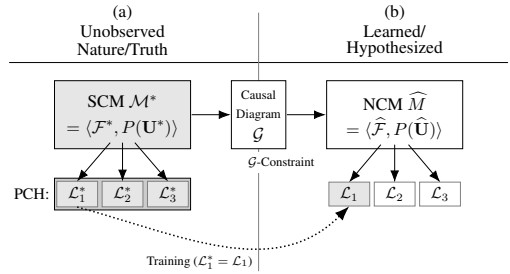

Figure 2: The l.h.s. contains the true SCM $\mathcal{M}^*$ that induces PCH's three layers. The r.h.s. contains an NCM that is trained with layer 1 data. The matching shading indicates that the two models agree with respect to $L_1$ while not necessarily agreeing in layers 2 and 3. The causal diagram $\mathcal{G}$ entailed by $\mathcal{M}^*$ is used as an inductive bias for $\widehat{M}$.

**Definition 7** ($\mathcal{G}$-Constrained NCM (constructive)). Let $\mathcal{G}$ be the causal diagram induced by SCM $\mathcal{M}^*$. Construct NCM $\widehat{M}$ as follows. **(1)** Choose $\widehat{\mathbf{U}}$ s.t. $\widehat{U}_{\mathbf{C}} \in \widehat{\mathbf{U}}$ if and only if $\mathbf{C}$ is a $C^2$-component in $\mathcal{G}$. **(2)** For each variable $V_i \in \mathbf{V}$, choose $\mathbf{Pa}_{V_i} \subseteq \mathbf{V}$ s.t. for every $V_j \in \mathbf{V}$, $V_j \in \mathbf{Pa}_{V_i}$ if and only if there is a directed edge from $V_j$ to $V_i$ in $\mathcal{G}$. Any NCM in this family is said to be $\mathcal{G}$-constrained. ∎

Note that this represents a family of NCMs, not a unique one, since $\boldsymbol{\theta}$ (the parameters of the neural networks) are not yet specified by the construction, only the scope of the function and independence relations among the sources of randomness ($\widehat{\mathbf{U}}$). In contrast to SCMs where both $\langle \mathcal{F}, P(\mathbf{u}) \rangle$ can freely vary, the degrees of freedom within NCMs come from $\boldsymbol{\theta}$. [7]

---

[6]Otherwise, the causal diagram can be learned through structural learning algorithms from observational data [65, 61] or experimental data [41, 40, 27]. See the next footnote for a neural take on this task.

[7]There is a growing literature that models SCMs using neural nets as functions, but which differ in nature and scope to our work. Broadly, these works assume Markovianity, which entails strong constraints over $P(U)$

We show next that an NCM constructed following the procedure dictated by Def. 7 encodes all the constraints of the original causal diagram.

**Theorem 2** (NCM $\mathcal{G}$-Consistency). *Any $\mathcal{G}$-constrained NCM $\widehat{M}(\boldsymbol{\theta})$ is $\mathcal{G}$-consistent.* ∎

We show next the implications of imposing the structural constraints embedded in the causal diagram.

**Theorem 3** ($L_2$-$\mathcal{G}$ Representation). *For any SCM $\mathcal{M}^*$ that induces causal diagram $\mathcal{G}$, there exists a $\mathcal{G}$-constrained NCM $\widehat{M}(\boldsymbol{\theta}) = \langle \widehat{\mathbf{U}}, \mathbf{V}, \widehat{\mathcal{F}}, P(\widehat{\mathbf{U}}) \rangle$ that is $L_2$-consistent w.r.t. $\mathcal{M}^*$.* ∎

The importance of this result stems from the fact that despite constraining the space of NCMs to those compatible with $\mathcal{G}$, the resultant family is still expressive enough to represent the entire Layer 2 of the original, unobserved SCM $\mathcal{M}^*$.

Fig. 2 provides a mental picture useful to understand the results discussed so far. The true SCM $\mathcal{M}^*$ generates the three layers of the causal hierarchy (left side), but in many settings only observational data (layer 1) is visible. An NCM $\widehat{M}$ trained with this data is capable of perfectly representing $L_1$ (right side). For almost any generating $\mathcal{M}^*$ sampled from the space $\Omega^*$, there exists an NCM $\widehat{M}$ that exhibits the same behavior with respect to observational data ($\widehat{M}$ is $L_1$-consistent) but exhibits a different behavior with respect to interventional data. In other words, $L_1$ underdetermines $L_2$. (Similarly, $L_1$ and $L_2$ underdetermine $L_3$ [5, Sec. 1.3].) Still, the true SCM $\mathcal{M}^*$ also induces a causal diagram $\mathcal{G}$ that encodes constraints over the interventional distributions. If we use this collection of constraints as an inductive bias, imposing $G$-consistency in the construction of the NCM, $\widehat{M}$ may agree with those of the true $\mathcal{M}^*$ under some conditions, which we will investigate in the next section.

## 3 The Neural Identification Problem

We now investigate the feasibility of causal inferences in the class of $\mathcal{G}$-constrained NCMs. [8] The first step is to refine the notion of identification [58, pp. 67] to inferences within this class of models.

**Definition 8** (Neural Effect Identification). Consider any arbitrary SCM $\mathcal{M}^*$ and the corresponding causal diagram $\mathcal{G}$ and observational distribution $P(\mathbf{V})$. The causal effect $P(\mathbf{y} \mid do(\mathbf{x}))$ is said to be neural-identifiable from the set of $\mathcal{G}$-constrained NCMs $\Omega(\mathcal{G})$ and observational distribution $P(\mathbf{V})$ if and only if $P^{\widehat{M}_1}(\mathbf{y} \mid do(\mathbf{x})) = P^{\widehat{M}_2}(\mathbf{y} \mid do(\mathbf{x}))$ for every pair of models $\widehat{M}_1, \widehat{M}_2 \in \Omega(\mathcal{G})$ s.t. $P^{\mathcal{M}^*}(\mathbf{V}) = P^{\widehat{M}_1}(\mathbf{V}) = P^{\widehat{M}_2}(\mathbf{V})$. ∎

In the context of graphical identifiability [58, Def. 3.2.4] and do-calculus, an effect is identifiable if any SCM in $\Omega^*$ compatible with the observed causal diagram and capable of generating the observational distribution matches the interventional query. If we constrain our attention to NCMs, identification in the general class would imply identification in NCMs, naturally, since it needs to hold for all SCMs. On the other hand, it may be insufficient to constrain identification within the NCM class, like in Def. 8, since it is conceivable that the effect could match within the class (perhaps in a not very expressive neural architecture) while there still exists an SCM that generates the same observational distribution and induces the same diagram, but does not agree in the interventional query; see Example 7 in Appendix C. The next result shows that this is never the case with NCMs, and there is no loss of generality when deciding identification through the NCM class.

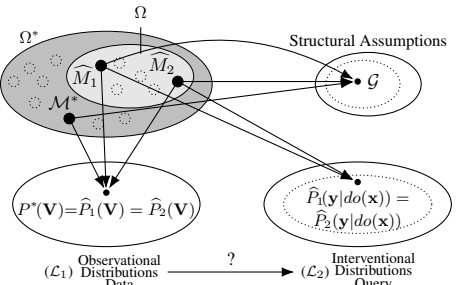

Figure 3: $P(\mathbf{y} \mid do(\mathbf{x}))$ is identifiable from $P(\mathbf{V})$ and $\Omega(\mathcal{G})$ if for any SCM $\mathcal{M}^* \in \Omega^*$ and NCMs $\widehat{M}_1, \widehat{M}_2 \in \Omega$ (top left), $\widehat{M}_1, \widehat{M}_2, \mathcal{M}^*$ match in $P(\mathbf{V})$ (bottom left) and $\mathcal{G}$ (top right), then the NCMs $\widehat{M}_1, \widehat{M}_2$ also match in $P(\mathbf{y} \mid do(\mathbf{x}))$ (bottom right).

---

and, in the context of identification, implies that all effects are always identifiable; see Corol. 3. For instance, [21] attempts to learn the entire SCM from observational ($L_1$) data, while [8, 10] also leverages experimental ($L_2$) data. On the inference side, [42] focuses on estimating causal effects of labels on images.

[8]This is akin to what happens with the non-neural CHT [5, Thm. 1] and the subsequent use of causal diagrams to encode the necessary inductive bias, and in which the do-calculus allows for cross-layer inferences directly from the graphical representation [5, Sec. 1.4].

**Theorem 4** (Graphical-Neural Equivalence (Dual ID)). *Let $\Omega^*$ be the set of all SCMs and $\Omega$ the set of NCMs. Consider the true SCM $\mathcal{M}^*$ and the corresponding causal diagram $\mathcal{G}$. Let $Q = P(\mathbf{y} \mid do(\mathbf{x}))$ be the query of interest and $P(\mathbf{V})$ the observational distribution. Then, $Q$ is neural identifiable from $\Omega(\mathcal{G})$ and $P(\mathbf{V})$ if and only if it is identifiable from $\mathcal{G}$ and $P(\mathbf{V})$.* ∎

In words, Theorem 4 relates the solution space of these two classes of models, which means that the identification status of a query is preserved across settings. For instance, if an effect is identifiable from the combination of a causal graph $\mathcal{G}$ and $P(\mathbf{v})$, it will also be identifiable from $\mathcal{G}$-constrained NCMs (and the other way around). This is encouraging since our goal is to perform inferences directly through neural causal models, within $\Omega(\mathcal{G})$, avoiding the symbolic nature of do-calculus computation; the theorem guarantees that this is achievable *in principle*.

**Corollary 2** (Neural Mutilation (Operational ID)). *Consider the true SCM $\mathcal{M}^* \in \Omega^*$, causal diagram $\mathcal{G}$, the observational distribution $P(\mathbf{V})$, and a target query $Q$ equal to $P^{\mathcal{M}^*}(\mathbf{y} \mid do(\mathbf{x}))$. Let $\widehat{\mathcal{M}} \in \Omega(\mathcal{G})$ be a $\mathcal{G}$-constrained NCM that is $L_1$-consistent with $\mathcal{M}^*$. If $Q$ is identifiable from $\mathcal{G}$ and $P(\mathbf{V})$, then $Q$ is computable through a mutilation process on a proxy NCM $\widehat{\mathcal{M}}$, i.e., for each $X \in \mathbf{X}$, replacing the equation $f_x$ with a constant $x$ ($Q = \text{PROC-MUTILATION}(\widehat{M}; \mathbf{X} = \mathbf{x}, \mathbf{Y})$).* ∎

Following the duality stated by Thm. 4, this result provides a practical, operational way of evaluating queries in NCMs: inferences may be carried out through the process of mutilation, which gives semantics to queries in the generating SCM $\mathcal{M}^*$ (via Def. 2). What is interesting here is that the proposition provides conditions under which this process leads to valid inferences, even when $\mathcal{M}^*$ is unknown, or when the mechanisms $\mathcal{F}$ and exogenous distribution $P(\mathbf{U})$ of $\mathcal{M}^*$ and the corresponding functions and distribution of the proxy NCM $\widehat{M}$ do not match. (For concreteness, refer to example 5 in Appendix. C.) In words, inferences using mutilation on $\widehat{M}$ would work as if they were on $\mathcal{M}^*$ itself, and they would be correct so long as certain stringent properties were satisfied – $L_1$-consistency, $\mathcal{G}$-constraint, and identifiability. As shown earlier, if these properties are not satisfied, inferences within a proxy model will almost never be valid, likely bearing no relationship with the ground truth. (For fully worked out instances of this situation, refer to examples 2, 3, or 4 in Appendix C).

Still, one special class of SCMs in which any interventional distribution is identifiable is called *Markovian*, where all $U_i$ are assumed independent and affect only one endogenous variable $V_i$.

**Corollary 3** (Markovian Identification). *Whenever the $\mathcal{G}$-constrained NCM $\widehat{\mathcal{M}}$ is Markovian, $P(\mathbf{y} \mid do(\mathbf{x}))$ is always identifiable through the process of mutilation in the proxy NCM (via Corol. 2).* ∎

This is obviously not the case for general non-Markovian models, which leads to the very problem of identification. In these cases, we need to decide whether the mutilation procedure (Corol. 2) can, in principle, produce the correct answer. We show in Alg. 1 a learning procedure that decides whether a certain effect is identifiable from observational data. Intuitively, the procedure searches for two models that respectively minimize and maximize the target query while maintaining $L_1$-consistency with the data distribution. If the $L_2$ query values induced by the two models are equal, then the effect is identifiable, and the value is returned; otherwise, the effect

---

**Algorithm 1**: Identifying/estimating queries with NCMs.

> **Input** : causal query $Q = P(\mathbf{y} \mid do(\mathbf{x}))$, $L_1$ data $P(\mathbf{V})$, and causal diagram $\mathcal{G}$
>
> **Output** : $P^{\mathcal{M}^*}(\mathbf{y} \mid do(\mathbf{x}))$ if identifiable, FAIL otherwise.

1  $\widehat{M} \leftarrow \text{NCM}(\mathbf{V}, \mathcal{G})$              // from Def. 7
2  $\boldsymbol{\theta}^*_{\min} \leftarrow \arg\min_{\boldsymbol{\theta}} P^{\widehat{M}(\boldsymbol{\theta})}(\mathbf{y} \mid do(\mathbf{x}))$ s.t. $L_1(\widehat{M}(\boldsymbol{\theta})) = P(\mathbf{V})$
3  $\boldsymbol{\theta}^*_{\max} \leftarrow \arg\max_{\boldsymbol{\theta}} P^{\widehat{M}(\boldsymbol{\theta})}(\mathbf{y} \mid do(\mathbf{x}))$ s.t. $L_1(\widehat{M}(\boldsymbol{\theta})) = P(\mathbf{V})$
4  **if** $P^{\widehat{M}(\boldsymbol{\theta}^*_{\min})}(\mathbf{y} \mid do(\mathbf{x})) \neq P^{\widehat{M}(\boldsymbol{\theta}^*_{\max})}(\mathbf{y} \mid do(\mathbf{x}))$ **then**
5  |    **return** FAIL
6  **else**
7  |    **return** $P^{\widehat{M}(\boldsymbol{\theta}^*_{\min})}(\mathbf{y} \mid do(\mathbf{x}))$    // choose min or max arbitrarily

---

is non-identifiable. Remarkably, the procedure is both necessary and sufficient, which means that all, and only, identifiable effects are classified as such by our procedure. This implies that, theoretically, deep learning could be as powerful as the do-calculus in deciding identifiability. (For a more nuanced discussion of symbolic versus optimization-based approaches for identification, see Appendix C.4. For non-identifiability examples and further discussion, see C.3.)

**Corollary 4** (Soundness and Completeness). *Let $\Omega^*$ be the set of all SCMs, $\mathcal{M}^* \in \Omega^*$ be the true SCM inducing causal diagram $\mathcal{G}$, $Q = P(\mathbf{y} \mid do(\mathbf{x}))$ be a query of interest, and $\widehat{Q}$ be the result from running Alg. 1 with inputs $P^*(\mathbf{V}) = L_1(\mathcal{M}^*) > 0$, $\mathcal{G}$, and $Q$. Then $Q$ is identifiable from $\mathcal{G}$ and $P^*(\mathbf{V})$ if and only if $\widehat{Q}$ is not FAIL. Moreover, if $\widehat{Q}$ is not FAIL, then $\widehat{Q} = P^{\mathcal{M}^*}(\mathbf{y} \mid do(\mathbf{x}))$.* ∎

# 4 The Neural Estimation Problem

While identifiability is fully solved by the asymptotic theory discussed so far (i.e., it is both necessary and sufficient), we now consider the problem of estimating causal effects in practice under imperfect optimization and finite samples and computation. For concreteness, we discuss next the discrete case with binary variables, but our construction extends naturally to categorical and continuous variables (see Appendix B). We propose next a construction of a $\mathcal{G}$-constrained NCM $\widehat{M}(\mathcal{G}; \boldsymbol{\theta}) = \langle \widehat{\mathbf{U}}, \mathbf{V}, \widehat{\mathcal{F}}, P(\widehat{\mathbf{U}}) \rangle$, which is a possible instantiation of Def. 7:

$$
\begin{cases}
\mathbf{V} & := \mathbf{V}, \ \widehat{\mathbf{U}} := \{U_{\mathbf{C}} : \mathbf{C} \in C^2(\mathcal{G})\} \cup \{G_{V_i} : V_i \in \mathbf{V}\}, \\
\widehat{\mathcal{F}} & := \left\{ f_{V_i} := \arg\max_{j \in \{0,1\}} g_{j,V_i} + \begin{cases} \log \sigma(\phi_{V_i}(\mathbf{pa}_{V_i}, \mathbf{u}_{V_i}^c; \theta_{V_i})) & j = 1 \\ \log(1 - \sigma(\phi_{V_i}(\mathbf{pa}_{V_i}, \mathbf{u}_{V_i}^c; \theta_{V_i}))) & j = 0 \end{cases} \right\}, \\
P(\widehat{\mathbf{U}}) & := \{U_{\mathbf{C}} \sim \text{Unif}(0,1) : U_{\mathbf{C}} \in \mathbf{U}\} \cup \\
& \quad \{G_{j,V_i} \sim \text{Gumbel}(0,1) : V_i \in \mathbf{V}, j \in \{0,1\}\},
\end{cases} \quad (2)
$$

where $\mathbf{V}$ are the nodes of $\mathcal{G}$; $\sigma : \mathbb{R} \to (0,1)$ is the sigmoid activation function; $C^2(\mathcal{G})$ is the set of $C^2$-components of $\mathcal{G}$; each $G_{j,V_i}$ is a standard Gumbel random variable [24]; each $\phi_{V_i}(\cdot; \theta_{V_i})$ is a neural net parameterized by $\theta_{V_i} \in \boldsymbol{\theta}$; $\mathbf{pa}_{V_i}$ are the values of the parents of $V_i$; and $\mathbf{u}_{V_i}^c$ are the values of $\mathbf{U}_{V_i}^c := \{U_{\mathbf{C}} : U_{\mathbf{C}} \in \mathbf{U} \text{ s.t. } V_i \in \mathbf{C}\}$. The parameters $\boldsymbol{\theta}$ are not yet specified and must be learned through training to enforce $L_1$-consistency (Def. 4).

Let $\mathbf{U}^c$ and $\mathbf{G}$ denote the latent $C^2$-component variables and Gumbel random variables, respectively. To estimate $P^{\widehat{M}}(\mathbf{v})$ and $P^{\widehat{M}}(\mathbf{y} \mid do(\mathbf{x}))$ given Eq. 2, we may compute the probability mass of a datapoint $\mathbf{v}$ with intervention $do(\mathbf{X} = \mathbf{x})$ ($\mathbf{X}$ is empty when observational) as:

$$
P^{\widehat{M}(\mathcal{G}; \boldsymbol{\theta})}(\mathbf{v} \mid do(\mathbf{x})) = \mathbb{E}_{P(\mathbf{u}^c)} \left[ \prod_{V_i \in \mathbf{V} \setminus \mathbf{X}} \tilde{\sigma}_{v_i} \right] \approx \frac{1}{m} \sum_{j=1}^{m} \prod_{V_i \in \mathbf{V} \setminus \mathbf{X}} \tilde{\sigma}_{v_i}, \quad (3)
$$

where $\tilde{\sigma}_{v_i} := \begin{cases} \sigma(\phi_i(\mathbf{pa}_{V_i}, \mathbf{u}_{V_i}^c; \theta_{V_i})) & v_i = 1 \\ 1 - \sigma(\phi_i(\mathbf{pa}_{V_i}, \mathbf{u}_{V_i}^c; \theta_{V_i})) & v_i = 0 \end{cases}$ and $\{\mathbf{u}_j^c\}_{j=1}^m$ are samples from $P(\mathbf{U}^c)$. Here, we assume $\mathbf{v}$ is consistent with $\mathbf{x}$ (the values of $X \in \mathbf{X}$ in $\mathbf{v}$ match the corresponding ones of $\mathbf{x}$). Otherwise, $P^{\widehat{M}(\mathcal{G}; \boldsymbol{\theta})}(\mathbf{v} \mid do(\mathbf{x})) = 0$. For numerical stability of each $\phi_i(\cdot)$, we work in log-space and use the log-sum-exp trick.

Alg. 1 (lines 2-3) requires non-trivial evaluations of expressions like $\arg\max_{\boldsymbol{\theta}} P^{\widehat{M}}(\mathbf{y} \mid do(\mathbf{x}))$ while enforcing $L_1$-consistency. Whenever only finite samples are available $\{\mathbf{v}_k\}_{k=1}^n \sim P^*(\mathbf{V})$, the parameters of an $L_1$-consistent NCM may be estimated by minimizing data negative log-likelihood:

$$
\boldsymbol{\theta} \in \arg\min_{\boldsymbol{\theta}} \mathbb{E}_{P^*(\mathbf{v})} \left[ -\log P^{\widehat{M}(\mathcal{G}; \boldsymbol{\theta})}(\mathbf{v}) \right]
$$

$$
\approx \arg\min_{\boldsymbol{\theta}} \frac{1}{n} \sum_{k=1}^{n} -\log \widehat{P}_m^{\widehat{M}(\mathcal{G}; \boldsymbol{\theta})}(\mathbf{v}_k). \quad (4)
$$

To simultaneously maximize $P^{\widehat{M}}(\mathbf{y} \mid do(\mathbf{x}))$, we subtract a weighted second term $\log \widehat{P}_m^{\widehat{M}}(\mathbf{y} \mid do(\mathbf{x}))$, resulting in the objective $\mathcal{L}(\{\mathbf{v}_k\}_{k=1}^n)$ equal to

$$
\frac{1}{n} \sum_{k=1}^{n} -\log \widehat{P}_m^{\widehat{M}}(\mathbf{v}_k) - \lambda \log \widehat{P}_m^{\widehat{M}}(\mathbf{y} \mid do(\mathbf{x})), \quad (5)
$$

where $\lambda$ is initially set to a high value and decreases during training. To minimize, we instead subtract $\lambda \log(1 - \widehat{P}_m^{\widehat{M}}(\mathbf{y} \mid do(\mathbf{x})))$ from the log-likelihood.

---

**Algorithm 2**: Training Model

**Input** : Data $\{\mathbf{v}_k\}_{k=1}^n$, variables $\mathbf{V}, \mathbf{X} \subseteq \mathbf{V}$, $\mathbf{x} \in \mathcal{D}_\mathbf{X}, \mathbf{Y} \subseteq \mathbf{V}, \mathbf{y} \in \mathcal{D}_\mathbf{Y}$, causal diagram $\mathcal{G}$, number of Monte Carlo samples $m$, regularization constant $\lambda$, learning rate $\eta$

1   $\widehat{M} \leftarrow \text{NCM}(\mathbf{V}, \mathcal{G})$     // from Def. 7
2   Initialize parameters $\boldsymbol{\theta}_{\min}$ and $\boldsymbol{\theta}_{\max}$
3   **for** $k \leftarrow 1$ **to** $n$ **do**
    // Estimate from Eq. 3
4     $\hat{p}_{\min} \leftarrow \text{Estimate}(\widehat{M}(\boldsymbol{\theta}_{\min}), \mathbf{V}, \mathbf{v}_k, \emptyset, \emptyset, m)$
5     $\hat{p}_{\max} \leftarrow \text{Estimate}(\widehat{M}(\boldsymbol{\theta}_{\max}), \mathbf{V}, \mathbf{v}_k, \emptyset, \emptyset, m)$
6     $\hat{q}_{\min} \leftarrow 0$
7     $\hat{q}_{\max} \leftarrow 0$
8     **for** $\mathbf{v} \in \mathcal{D}_\mathbf{V}$ **do**
9       **if** $\text{Consistent}(\mathbf{v}, \mathbf{y})$ **then**
10        $\hat{q}_{\min} \leftarrow \hat{q}_{\min} +$ $\text{Estimate}(\widehat{M}(\boldsymbol{\theta}_{\min}), \mathbf{V}, \mathbf{v}, \mathbf{X}, \mathbf{x}, m)$
11        $\hat{q}_{\max} \leftarrow \hat{q}_{\max} +$ $\text{Estimate}(\widehat{M}(\boldsymbol{\theta}_{\max}), \mathbf{V}, \mathbf{v}, \mathbf{X}, \mathbf{x}, m)$
    // $\mathcal{L}$ from Eq. 5
12     $\mathcal{L}_{\min} \leftarrow -\log \hat{p}_{\min} - \lambda \log(1 - \hat{q}_{\min})$
13     $\mathcal{L}_{\max} \leftarrow -\log \hat{p}_{\max} - \lambda \log \hat{q}_{\max}$
14     $\boldsymbol{\theta}_{\min} \leftarrow \boldsymbol{\theta}_{\min} + \eta \nabla \mathcal{L}_{\min}$
15     $\boldsymbol{\theta}_{\max} \leftarrow \boldsymbol{\theta}_{\max} + \eta \nabla \mathcal{L}_{\max}$

---

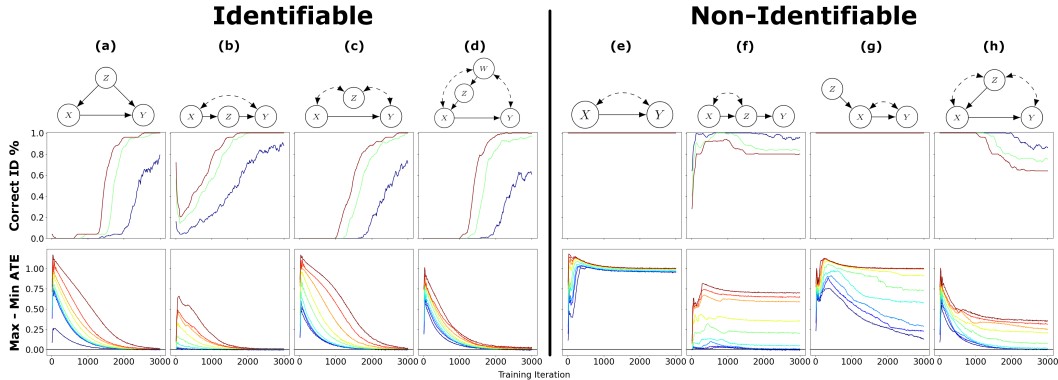

Figure 4: Experimental results on deciding identifiability with NCMs. **Top**: Graphs from left to right: (ID cases) back-door, front-door, M, napkin; (not ID cases) bow, extended bow, IV, bad M. **Middle**: Classification accuracy over 3,000 training epochs from running hypothesis test on Eq. 6 with $\tau = 0.01$ (blue), 0.03 (green), 0.05 (red). **Bottom**: (1, 5, 10, 25, 50, 75, 90, 95, 99)-percentiles for max-min gaps over 3000 training epochs.

Alg. 2 is one possible way of optimizing the parameters $\boldsymbol{\theta}$ required in lines 2,3 of Alg. 1. Eq. 5 is amenable to optimization through standard gradient descent tools, e.g., [38, 51, 50]. [9] [10]

One way of understanding Alg. 1 is as a search within the $\Omega(\mathcal{G})$ space for two NCM parameterizations, $\boldsymbol{\theta}_{\min}^*$ and $\boldsymbol{\theta}_{\max}^*$, that minimizes/maximizes the interventional distribution, respectively. Whenever the optimization ends, we can compare the corresponding $P(\mathbf{y} \mid do(\mathbf{x}))$ and determine whether an effect is identifiable. With perfect optimization and unbounded resources, identifiability entails the equality between these two quantities. In practice, we rely on a hypothesis testing step such as

$$|f(\widehat{M}(\boldsymbol{\theta}_{\max})) - f(\widehat{M}(\boldsymbol{\theta}_{\min}))| < \tau \tag{6}$$

for quantity of interest $f$ and a certain threshold $\tau$. This threshold is somewhat similar to a significance level in statistics and can be used to control certain types of errors. In our case, the threshold $\tau$ can be determined empirically. For further discussion, see Appendix B.

## 5    Experiments

We start by evaluating NCMs (following Eq. 2) in their ability to decide whether an effect is identifiable through Alg. 2. Observational data is generated from 8 different SCMs, and their corresponding causal diagrams are shown in Fig. 4 (top part), and Appendix B provides further details of the parametrizations. Since the NCM does not have access to the true SCM, the causal diagram and generated datasets are passed to the algorithm to decide whether an effect is identifiable. The target effect is $P(Y \mid do(X))$, and the quantity we optimize is the *average treatment effect* (ATE) of $X$ on $Y$, $ATE_{\mathcal{M}}(X, Y) = \mathbb{E}_{\mathcal{M}}[Y \mid do(X = 1)] - \mathbb{E}_{\mathcal{M}}[Y \mid do(X = 0)]$. Note that if the outcome $Y$ is binary, as in our examples, $\mathbb{E}[Y \mid do(X = x)] = P(Y = 1 | do(X = x))$. The effect is identifiable through do-calculus in the settings represented by Fig. 4 in the left part, and not identifiable in right.

The bottom row of Fig. 4 shows the *max-min gaps*, the l.h.s of Eq. 6 with $f(\mathcal{M}) = ATE_{\mathcal{M}}(X, Y)$, over 3000 training epochs. The parameter $\lambda$ is set to 1 at the beginning, and decreases logarithmically over each epoch until it reaches 0.001 at the end of training. The max-min gaps can be used to classify the quantity as "ID" or "non-ID" using the hypothesis testing procedure described in Appendix B. The classification accuracies per training epoch are shown in Fig. 4 (middle row). Note that in identifiable settings, the gaps slowly reduce to 0, while the gaps rapidly grow and stay high throughout training in the unidentifiable ones. The classification accuracy for ID cases then gradually increases as training

---

[9]Our approach is flexible and may take advantage of these different methods depending on the context. There are a number of alternatives for minimizing the discrepancy between $P^*$ and $P^{\widehat{M}}$, including minimizing divergences, such as maximum mean discrepancy [23] or kernelized Stein discrepancy [49], performing variational inference [9], or generative adversarial optimization [20].

[10]The NCM can be extended to the continuous case by replacing the Gumbel-max trick on $\sigma(\phi_i(\cdot))$ with a model that directly computes a probability density given a data point, e.g., normalizing flow [62] or VAE [39].

progresses, while accuracy for non-ID cases remain high the entire time (perfect in the bow and IV cases).

In the identifiable settings, we also evaluate the performance of the NCM at estimating the correct causal effect, as shown in Fig. 5. As a generative model, the NCM is capable of generating samples from both $P(\mathbf{V})$ and identifiable $L_2$ distributions like $P(Y \mid do(X))$. We compare the NCM to a naïve generative model trained via likelihood maximization fitted on $P(\mathbf{V})$ without using the inductive bias of the NCM. Since the naïve model is not defined to sample from $P(y \mid do(x))$, this shows the implications of arbitrarily choosing $P(y \mid do(x)) = P(y \mid x)$. Both models improve at fitting $P(\mathbf{V})$ with more samples, but the naïve model fails to learn the correct ATE except in case (c), where $P(y \mid do(x)) = P(y \mid x)$. Further, the NCM is competitive with WERM [33], a state-of-the-art estimation method that directly targets estimating the causal effect without generating samples.

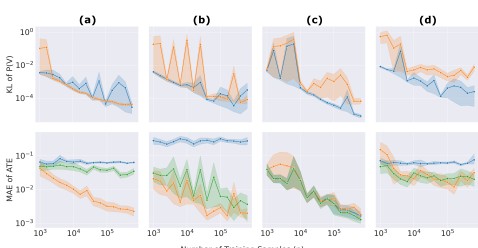

Figure 5: NCM estimation results for ID cases. Columns a, b, c, d correspond to the same graphs as a, b, c, d in Fig. 4. **Top**: KL divergence of $P(\mathbf{V})$ induced by naïve model (blue) and NCM (orange) compared to $P^{\mathcal{M}^*}(\mathbf{V})$. **Bottom**: MAE of ATE of naïve model (blue), NCM (orange), and WERM (green). Plots in log-log scale.

## 6    Conclusions

In this paper, we introduced neural causal models (NCMs) (Def. 3, 18), a special class of SCMs trainable through gradient-based optimization techniques. We showed that despite being as expressive as SCMs (Thm. 1), NCMs are unable to perform cross-layer inferences in general (Corol. 1). Disentangling expressivity and learnability, we formalized a new type of inductive bias based on non-parametric, structural properties of the generating SCM, accompanied with a constructive procedure that allows NCMs to represent constraints over the space of interventional distributions akin to causal diagrams (Thm. 2). We showed that NCMs with this bias retain their full expressivity (Thm. 3) but are now empowered to solve canonical tasks in causal inference, including the problems of identification and estimation (Thm. 4). We grounded these results by providing a training procedure that is both sound and complete (Alg. 1, 2, Cor. 4). Practically speaking, different neural implementations – combination of architectures, training algorithms, loss functions – can leverage the framework results introduced in this work (Appendix D.1). We implemented one of such alternatives as a proof of concept, and experimental results support the feasibility of the proposed approach. After all, we hope the causal-neural framework established in this paper can help develop more principled and robust architectures to empower the next generation of AI systems. We expect these systems to combine the best of both worlds by (1) leveraging causal inference capabilities of processing the structural invariances found in nature to construct more explainable and generalizable decision-making procedures, and (2) leveraging deep learning capabilities to scale inferences to handle challenging, high dimensional settings found in practice.

## Acknowledgements

We thank Judea Pearl, Richard Zemel, Yotam Alexander, Juan Correa, Sanghack Lee, and Junzhe Zhang for their valuable feedback. Kevin Xia and Elias Bareinboim were supported in part by funding from the NSF, Amazon, JP Morgan, and The Alfred P. Sloan Foundation. Yoshua Bengio was supported in part by funding from CIFAR, NSERC, Samsung, and Microsoft.

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
