# OpenReview forum: "The Causal-Neural Connection: Expressiveness, Learnability, and Inference"
_NeurIPS.cc/2021/Conference — NeurIPS 2021 Poster_

### Official Review · Reviewer_fTRd · 2021-07-15

**Rating:** 6
**Confidence:** 2

**Summary:**

In this paper a subclass of SCMs whose structural equations are based on feed-forward NNs are considered.
The authors show the expressiveness of this class and address the problem of training those model and evaluated identifiability of a causal task. Experiments on models (with few variables) are promising.

**Ethical Concerns:**

Nothing.

**Limitations And Societal Impact:**

Nothing.

**Main Review:**

Causal machine learning is definitely an important field of research for the NeurIPS community. This paper offers an interesting class of models to practically achieve that. Considering that NNs are universal approximators, the results in the technical part are not surprising, but I think the paper might really help researchers in this area. The fact that the experiments are only focused on very small models leave some doubts about the practical scalability of these techniques.

**Time Spent Reviewing:**

1

---

> ### Author Response · Authors · 2021-08-10
> **Response to Reviewer fTRd**
>
> We thank the reviewer for all the comments and feedback provided. Please find our responses next.
>
> ***
> > _Considering that NNs are universal approximators, the results in the technical part are not surprising (...)_
> ***
>
> **Reply:**
> Following this observation, we would like to respectfully point out what we believe is a gap in understanding of our contributions leading to the somewhat negative evaluation, and we would like to provide further clarification. One important motivator for the investigation in this paper is that universal approximation of the conditionals is not sufficient to obtain identifiability of the causal effect and thus proper generalization to new interventions, which motivates the bulk of our contributions. More specifically, the paper is underpinned by the following logic:
>
>  (1) It is the case that NNs are universal approximators (UAs), and, therefore, if we construct NCMs using a collection of NNs, they will also exhibit the property of universal approximability.
>
>  (2) Departing from this observation, we ask whether universal approximability could immediately solve a typical causal inference task.
>   For instance, we consider the canonical problem of identifying the effect of new interventions from observational data, which is the topic of Ch. 3 in Causality (Pearl, 2000).
>
>  (3) We note that *despite* the UA, the NCMs constructed with NNs are unable to perform such tasks. This encompasses all the discussion culminating in Corollary 1 (beginning of Sec. 2).
>  We believe this step helps to “disentangle” and clarify that the UA property and the assumptions required to perform causal inferences are distinct; one does not imply the other, and they are both needed.
>
> (4) The remainder of the paper (from Sec. 2.1 onwards) is then trying to endow NNs (which are certainly UAs) with causal inductive biases that would allow them to solve causal inference tasks from first principles.
>
> All in all, we then respectfully note that this is not obvious at all, since the obvious does not quite work in this case.
>
> ***
> > _The fact that the experiments are only focused on very small models leaves some doubts about the practical scalability of these techniques._
> ***
>
> **Reply:**
> One of our motivations for this work is to leverage the scalability of neural models, which seems to work well in practice in a number of compelling scenarios, as noted in lines 90-92:
>  “It is our goal here to develop a general causal-neural framework that has the potential to scale to real-world, high-dimensional data domains while preserving the validity of the inferences like the traditional, symbolic approaches.”
>
> We note, however, that there are two parts of the statement -- leverage NNs scalability *while* still preserving the validity of the causal claims made by these inferences. Achieving both in general settings was not possible before this paper. Once the foundations are right, in the sense described in the previous discussion, we can perform causal inferences using neural models with guarantees, and the question of scalability arises. We indeed performed some preliminary experiments with settings up to 20 covariates, as shown in Fig. 8, Appendix D, which seem to corroborate with our approach. Concretely, the results are competitive with and/or superior to the current SOTA, which is usually what is required from any paper. Still, having said that, we should mention that we feel this work is not the last, but the first step towards having a solid theory and methods for neural-causal inferences, and we do expect a lot of future experimentation with different architectures, widths, depths, and other parameterizations that could lead to more efficient, large scale causal inferences. Finally, we believe that our empirical and theoretical contributions are in the right place for these developments to take place. We will certainly release the code of the paper to encourage other experienced deep learning researchers to test the limits of the NCM.

---

### Official Review · Reviewer_Amv3 · 2021-07-16

**Rating:** 7
**Confidence:** 3

**Summary:**

The paper considers a restricted class of SCMs (structural causal models) where each variable in a graph is generated by applying a neural-network-parameterized function of the variable's parents, called Neural Causal Models (NCMs). The paper shows a wealth of results mapping the relationship between the full class of SCMs and the smaller class of NCMs in terms of when and how identification holds and how estimation of the interventional distribution can be achieved. The method proposed to identify and estimate effects takes a graph as an input, computes the maximum and minimum of the effect of a particular intervention within the class of NCMs that are consistent with the observed data distribution (at the lowest level in the causal hierarchy, observational); if the maximum and minimum match, the effect is identified. The method is evaluated on a few experiments.

**Limitations And Societal Impact:**

I am concerned about the applicability of the proposed method to estimation of interventional distributions in general seeing that the experiments are limited. It is unclear why WERM is the only (non-naive) method compared against and how it is state-of-art. If the authors could explain why no other baselines were considered, that would help clarify the standing of the paper; for example, compare against interventional distributions using knowledge of the graph and effect estimation methods like like https://arxiv.org/abs/2102.12034 and https://arxiv.org/abs/2001.07426.

**Main Review:**

The paper is well-written and puts forth an interesting perspective of estimating interventional distributions. However, it is unclear why the notion of neural networks is central here in theory. I understand that practically there are appealing properties in terms of optimization and techniques used to quickly build good predictors using neural networks. However, all of the theory about NCMs being an interesting class of causal models, i.e. with identification and estimation, holds for any sufficiently flexible class of functions (lemmas 4,5 are constructive and nice but any large enough function class contains the constructed functions).

I think the distinction above is important because under this perspective, the main claim of the paper is that neural networks help learn interventional distributions better than other existing causal effect estimation methods that do not rely on the machinery of the NCMs; for example, one could check causal identifiability based on the given graph and then apply causal effect estimation methods like the following: like https://arxiv.org/abs/2102.12034 and https://arxiv.org/abs/2001.07426.

As the theory holds for any flexible class of functions, it seems to me that the experiments should show that the proposed estimation method is better than other baseline methods as suggested above.  As these baselines are not included, the experiments, while encouraging, are insufficient to establish the improvement that the proposed method (and NCM-based methodology) brings to causal effect estimation.


--- UPDATES after rebuttal

score updated after clarifying the position of the paper.

**Time Spent Reviewing:**

4

---

> ### Author Response · Authors · 2021-08-10
> **Response to Reviewer Amv3**
>
> We thank the reviewer for the detailed comments and valuable feedback. We feel that a few misreadings of our paper make the evaluation a bit harsh, and hope you can reconsider the paper based on the clarifications provided below.
>
> ***
> > _The paper is well-written and puts forth an interesting perspective of estimating interventional distributions. However, it is unclear why the notion of neural networks is central here in theory. (...)_
> ***
>
> **Reply:**
> We note that while the central results of this work are framed in the context of neural models, there is nothing preventing one from achieving the same results using a model constructed using some other expressive function class. In fact, we addressed this issue in Appendix D (starting from p. 38) by providing Def. 18, a generalized definition of the NCM where the functions do not necessarily have to be neural networks. This definition is paired with Thms. 5 and 6, which generalize Thms. 1 and 3. In the body of the paper, we focused on the definition using feedforward neural networks and uniform noise for the sake of clarity and exposition. Specifically, Secs. 4 and 5 provide a concrete model to ground the discussion on how to solve the identification and estimation problems in practice. Our goal is certainly not to isolate the class of neural networks specifically and exclude others. We will indeed point out this observation earlier on in the paper rather than in a footnote and throughout the appendices; thank you for the suggestion.
>
> ***
> > _one could check causal identifiability based on the given graph and then apply causal effect estimation methods like the following: like https://arxiv.org/abs/2102.12034 and https://arxiv.org/abs/2001.07426 (...) the experiments should show that the proposed estimation method is better than other baseline methods as suggested above. (...)_
> ***
>
> **Reply:**
> We appreciate the references and will cite accordingly. Regarding the specific suggestion, these two papers seem to propose interesting solutions, but we believe their scope differs significantly from ours. Notably, Johansson et al. assumes conditional ignorability in assumption 1, and assumption 3 in Kennedy et al. in Sec. 2 is equivalent to conditional ignorability (also known as back-door admissibility): given treatment $X$, covariates $Z$, and outcome $Y$, with $Y_{0}$ and $Y_{1}$ denoting $Y$ under the interventions $do(X=0)$ and $do(X=1)$, respectively, we have
>
> $$X \perp \{Y_0, Y_1\} | Z.$$
>
> Conditional ignorability may be a valid assumption in some settings but the method provided in our paper does not make or require it a priori. In fact, there are many settings to which our method can be applied and these others cannot. For example, in the four identifiable cases we present in Fig. 4, these assumptions only hold in graph (a), the back-door graph. These two methods are therefore not applicable to performing estimation in the other three identifiable cases (Fig. 4(b-d)). Since the NCM can handle arbitrary causal diagrams, we believe that it is fair to compare it against other methods that can handle the same space of models. Still, we were not aware of these works and appreciate you sharing them; we will add them in the citations around the more comparable models (lines 83-84).
>
> More generally, the solutions we introduce with the NCM solve the problems of effect identification and estimation under any arbitrary causal diagram. Our experiments in Figs. 5 and 8 provide empirical evidence that indicates our method matches or improves upon WERM in estimating causal effects among a variety of causal diagrams. To the best of our knowledge, WERM is the only method that can efficiently solve the problem of causal effect identification and estimation in the aforementioned general-graph setting. Thus, we use WERM as our baseline.
>
> ***
> > _the main claim of the paper is that neural networks help learn interventional distributions better than other existing causal effect estimation methods that do not rely on the machinery of the NCMs_
> ***
>
> **Reply:**
> For the reasons described above, the solution to the identification task is a vital contribution of our work, in addition to a generic neural method for doing estimation in arbitrary causal settings (when ignorability does not necessarily hold). In other words, the paper solves two tasks, identification and estimation. As established in Corol. 1, naively using a neural model without proper assumptions will result in incorrect causal inferences in the learned model, regardless of the sample size. The causal inductive bias introduced in our work - akin to the non-neural counterpart, causal diagrams & the do-calculus - bridges the gap between observational and interventional quantities, and identification is needed to decide whether certain interventional quantities can be inferred from the inputted observational distribution. The papers cited above do not need to consider the identification task since the target interventional distribution is guaranteed to be identifiable under the ignorability assumption, which is not assumed a priori in the NCM framework. Given that we do not know whether the effect is identifiable in the general setting, the NCM is designed in such a way that it can determine whether the quantity is or is not inferred from the combination of the observational data and causal inductive biases. Whenever the effect is identifiable, we can go to the second task of estimating the interventional distribution properly. We elaborate further in Appendix C.4 why we believe neural identification is a powerful contribution, in addition to neural estimation of causal effects under general settings.
>
> This is certainly an interesting discussion that we expect to highlight in the paper. We hope that the issues raised were clarified based on this discussion, but we will certainly be happy to provide further elaborations if you find suitable.

---

> > ### Comment · Reviewer_Amv3 · 2021-08-25
> > **Thanks for the clarification.**
> >
> > The authors clarified a bunch of things about the paper for me. I'm updating the score accordingly.
> >
> > I do believe the proposed method has a lot of experimental verification to go through before it can become a standard tool (which I believe it has the potential for) but that is not a burden on the authors now.

---

### Official Review · Reviewer_Z2BF · 2021-07-16

**Rating:** 8
**Confidence:** 2

**Summary:**

A new class of models called Neural Causal Models is proposed which retains causal reasoning properties of Structural Causal Models but can be learned through neural net based methods. contributions include formally proving connections between SCMs and NCMs, disentangling expressiveness with learnability of NCMs and deriving constrained classes of NCMs, a practical learning algorithm for NCMs.

**Limitations And Societal Impact:**

Societal impact not directly addressed, but since the paper addresses fundamental connections between symbolic and neural models, I cannot think of any negative societal impact or limitation to be concerned about.

**Main Review:**

The paper proposes a new model called Neural Causal model that has the properties similar to a structural causal model but can be learned through gradient-based methods. NCMs are a subset of SCMs but have causal reasoning power like SCMs. Formal properties are proved about the expressive power of NCMs, i.e., they can represent distributions as an SCM. However, this expressiveness does not mean that NCMs are learnable and require additional constraints imposed on the causal diagram. The constraints give rise to a learnable class of NCMs. It is shown that in this class it is possible to perform causal reasoning using observational data. Soundness and completeness results are presented for neural inference in NCMs and it is shown that it is equivalent to symbolic inference in SCMs. A gradient-based training method is presented for approximately learning NCMs.

Experiments are performed that show if an effect can be identified from a cause accurately through NCMs for different causal settings. For non-identifiable effects it is shown that the effects cannot be accurately identified through NCM learning. A generative model through NCM is compared with a generative model that does not use the causal graph (only data-driven) and also WERM, a state-of-the-art causal learning model.

The paper is well written and covers a lot given the space limits. The clarity of writing makes it accessible even to non-experts in this area I think. The significance of this work seems quite remarkable since it proves several fundamental connections between SCMs and neural nets. I can see that this work can have impact in many other domains such as explainability in AI. A case study of a real-world application would have been beneficial for understanding the impact in a practical setting. Overall, though this seems to be a highly significant paper connecting Neural and Symbolic AI.



**Time Spent Reviewing:**

5

---

> ### Author Response · Authors · 2021-08-10
> **Response to Reviewer Z2BF**
>
> We thank the reviewer for the positive assessment and encouraging comments. We were glad the work was understood and the fundamental connections between neural and symbolic causal inference methods became more clear through our work. We hope that the results we provide will serve as an initial motivation, or proof of concept, for practitioners from different fields interested in implementing a neural approach to causal inference and improving over what we proposed in this paper.

---

### Official Review · Reviewer_kLWB · 2021-07-20

**Rating:** 7
**Confidence:** 3

**Summary:**

This paper studies the expressivity, identification, estimation of causality with neural models. For expressiveness, it shows that Neural Causal Model (NCM) is expressive enough to represent any structural causal model (SCM), but in general, given observation data alone, without further constraints, the NCM is not able to identify higher layers of Pearl Causal Hierarchy (PCH). The authors then develop a necessary and sufficient condition to determine whether a causal effect can be learned from data (i.e., causal identifiability); if identifiability holds, it also estimates the effect from data (causal estimation). Empirical experiments on a simple synthetic dataset support the theoretical results.

**Limitations And Societal Impact:**

The authors did not address the limitation of the paper.

**Main Review:**

In terms of novelty, to my knowledge, the theory and the algorithms introduced are novel.

In terms of quality, I did not check through the theoretical proof in the Appendix, but I believe they are correct. For the experiments, they are evaluate on simple scenarios which demonstrate the effectiveness of the algorithm's capability of causal identification and estimation. I wonder how well the algorithm is able to scale to more complex scenarios where the observed number of variables and their edges are larger, and its run-time compared to training a standard network of similar structure. Furthermore, for the neural architecture, it would offer more insight if the authors can perform experiments that test if and how the expressivity of the neural models (number of layers, number of neurons per hidden layer, etc.) affects the results, and how many examples are needed for confident identification and estimation.

In terms of clarity, the paper is very theory-heavy. To improve intuition, if possible, it may be useful to use a running example, starting from the beginning, to accompany the proved theorems, so as to make it more friendly for readers not in the causal community. For example, the examples in Appendix C may be good examples.

Overall, I think this paper is well-written and of high-quality, and makes useful contribution to causal identification and estimation with neural methods. If the more thorough experiments are performed, it will further strengthen the paper.

**Time Spent Reviewing:**

5 hours

---

> ### Author Response · Authors · 2021-08-10
> **Response to Reviewer kLWB**
>
> We appreciate the reviewer for the positive review and encouraging comments, thank you. Please find our responses below.
>
> ***
> > _I wonder how well the algorithm is able to scale to more complex scenarios where the observed number of variables and their edges are larger, and its run-time compared to training a standard network of similar structure. Furthermore, for the neural architecture, it would offer more insight if the authors can perform experiments that test if and how the expressivity of the neural models (number of layers, number of neurons per hidden layer, etc.) affects the results, and how many examples are needed for confident identification and estimation._
> ***
>
> **Reply:**
> These are great suggestions, and we are interested in the results of performing these experiments as well. We are currently trying to obtain more computational resources to test NCMs in larger settings. Although we were only able to test some properties of one architecture choice, we believe that the provided results establish some empirical evidence that the algorithms we developed work as intended in practice. We will certainly release the code of the paper to encourage other experienced deep learning researchers to explore different NCM architectures and their properties.
>
> ***
> > _To improve intuition, if possible, it may be useful to use a running example, starting from the beginning, to accompany the proved theorems, so as to make it more friendly for readers not in the causal community. For example, the examples in Appendix C may be good examples_
> ***
>
> **Reply:**
> Certainly, thank you. It is indeed our goal and hope that readers from all communities can understand the results in this paper. The examples in Appendix C are referenced in the main body, but due to space constraints, we could not fit them directly into the text. Each example aims to provide intuition for one of the theorems or corollaries in the main text, and the setting modeled by the true SCM is shared across examples. In the case that this paper is accepted, we will use the extra page to improve the presentation and add concreteness through more examples, perhaps bringing some of them from Appendix C.

---

### Official Review · Reviewer_ue4r · 2021-07-21

**Rating:** 6
**Confidence:** 3

**Summary:**

The authors investigate the connection between structural causal models and neural networks from an learning perspective. Their theoretical results emphasize, notably, that the impossibility to learn neural networks that are interventionally equivalent to the true SCM based on observations only. The paper provides an interesting perspective on the connection between causal inference and learning with neural network, although clarity could be improved, notably regarding the limitations and the contribution with respect to previous work.

**Limitations And Societal Impact:**

Assumptions of finite domain seems to be central for the proofs, but does not seem to be mentioned in main text, while only a countability assumption is mentioned in only one lemma in supplemental, although finite domain seems to be exploited in the proof.

**Main Review:**

While the overall effort of this paper is clearly valuable, I struggled with the presentation of the theoretical results. It might be partly due to my partial understanding of the framework, although I think the main text lacks some key information.
1/ First, only by going through the appendix one can notice that the theory applies to variables in countable numerical domains (lemma 4). Besides, I tend to think this is not fully accurate, as the proof of lemma 4 seems to assume *finite* domains.
While one can argue that binary/finite domains can approximate continuous ones with arbitrary precision, it is important to specify it in the main text, as results would appear very confusing to readers familiar with technical issues, while unfamiliar readers who see the use of neural networks would naturally assume every functions defined can act on, say, Euclidean domains.
2/ It is confusing to have theorem 1 on L3 consistency while this property is not defined in main text. An informal presentation of Li would help.
3/ Corollary 1 conveys an important point, but only to those not familiar with causal inference (pointing to structural identifiability issues), for whom the level of abstraction will be difficult to parse. On the other hand, I tend to think that the setting of this result is too “coarse” to capture when causal inference is useful (and I assume the authors do agree with me that sometimes it is). If I did not miss anything, if we restrict ourselves to an SCM with two nodes, and we add additional information on the causal graph (e.g. uncounfounded, and the direction of causation in known), then we are in good shape to be able to collapse the interventional level to the observational one. This kind of positive result is not captured by the corollary one because a) we consider “all SCMs” at once, thus ignoring the informativeness of certain graph structures, b) we do not include the possibility to incorporate assumptions on the graph structure at inference, which is key in practice. I do get that the next sections address this issue, but I question the benefit this side result. Overall, I tend to think the target readers would benefit more from a high-level discussion, possibly in relation to the examples in supplemental, that from such abstract result (besides largely not self-contained in the paper).
4/ Overall, it is difficult to disentangle the contribution of the paper with respect to what is already known regarding learnability of SCMs. Once we have established the ability of neural networks to perfectly fit functions with finite domains (which can be found in the appendix, and are typically not too challenging to show), classical result from causality seem to apply as usual. I certainly may have missed key novel results that are fundamentally different from previous work, but they are not really emphasized in main text. It looks like the lemmas in appendix reflect better the gist of the contribution than the main text results.


**Time Spent Reviewing:**

4 hours

---

> ### Author Response · Authors · 2021-08-10
> **Response to Reviewer ue4r**
>
> We thank the reviewer for the detailed comments and valuable feedback. Please find our responses next.
>
> ***
> >  _1/ First, only by going through the appendix one can notice that the theory applies to variables in countable numerical domains (lemma 4). Besides, I tend to think this is not fully accurate, as the proof of lemma 4 seems to assume finite domains. While one can argue that binary/finite domains can approximate continuous ones with arbitrary precision, it is important to specify it in the main text (...)_
> ***
>
> **Reply:**
> Yes, the theory assumes we are working with SCMs where variables in V have finite domains. We made this assumption because some of the significant works that we extend, such as Bareinboim et al. (2020), also make a similar assumption, and as you point out, it is natural to reason that finite domains can approximate continuous domains in the limit. Since this assumption mainly affects Lem. 4 (and consequently Thms. 1 and 3), it is mostly a limitation to the expressivity results of the NCM. Perhaps one could prove results even stronger than Thms. 1 and 3 without the limitations, but we decided to instead prioritize the presentation of the impossibility result highlighted by Corol. 1 in the main body of the paper, which motivates the need for the causal diagram and corresponding inductive bias and is a more surprising result in our opinion. Having said that, we agree that mentioning this in the main body will clarify the scope of the theory; thank you for your suggestion.
>
> ***
> > _2/ It is confusing to have theorem 1 on L3 consistency while this property is not defined in main text. An informal presentation of Li would help._
> ***
>
> **Reply:**
> For the sake of clarity, this is indeed Def. 7 (p. 16) in Bareinboim et al., 2020, which is a bit more general and explicit than Eq. (7.4) in Causality (p. 206, Pearl, 2000).  Still, we will incorporate your suggestion around Def. 2, thanks.
>
> ***
> > _3/ Corollary 1 conveys an important point, but only to those not familiar with causal inference (pointing to structural identifiability issues), for whom the level of abstraction will be difficult to parse. On the other hand, I tend to think that the setting of this result is too “coarse” to capture when causal inference is useful (...)_
> ***
>
> **Reply:**
> Corol. 1 is intended to be a negative result, so it is “coarse” in the sense that it does *not* indicate settings where causal inferences can be accomplished, which includes the cases where the causal diagram is given. However, we believe it is an important result nonetheless. Without Corol. 1, non-causal researchers may be tempted to incorrectly believe that learning causal information from observational data can be possible in the general case without causal assumptions, as long as enough computational resources and data are provided. We use Corol. 1 to draw a line and emphasize that this is not the case, despite the universal expressivity property of the NCM. We aim to motivate the reader to consider (a) what kinds of assumptions or additional information may be needed to perform causal inference, and (b) how to decide when a certain causal quantity is identifiable in these settings. The impossibility delineated by Corol. 1 leads to a positive line of reasoning through the introduction of the causal diagram assumption and inductive bias, and the identifiability problem for (a) and (b), respectively. We hope that future work in causal inference will continue to consider these two points. The sentences around Corol. 1 largely try to capture this sentiment.  However, for improving clarity, we will add the following text to the end of line 178 to connect this negative result to the positive results described in later sections - thank you for your suggestion:
>
> "Thus, causal assumptions are necessary to perform causal inferences when using NCMs, despite their expressiveness. We discuss how to incorporate the necessary causal assumptions into an NCM to work around the limitation posed by Corol. 1 in the next subsection."
>
> ***
> > _4/ Overall, it is difficult to disentangle the contribution of the paper with respect to what is already known regarding learnability of SCMs. (...)_
> ***
>
> **Reply:**
> To provide a big-picture explanation on the implications of the results of this work, many previous works which solve the problems of causal identification and estimation operate on a level of abstraction without the SCM. They typically use the constraints of the causal diagram to symbolically derive a solution (e.g., do-calculus). Instead, this work introduces a framework where one can construct a proxy model to mimic the true SCM (on identifiable quantities) via neural optimization. Thm. 4 and Corols. 2, 4 indicate that identification under our particular proxy model specification, the NCM, is equivalent to symbolic identification through do-calculus. This is a key result in demonstrating that the NCM may be used as a proxy for the true SCM to yield a solution to the problems of causal identification and estimation, the main contribution of our work.
>
> The use of a proxy model is remarkable especially as a solution to the identification problem, because it shows that a data-driven approach with neural networks can perform causal reasoning, which had previously only been accomplished symbolically (e.g., through Pearl’s do-calculus). There are additional reasons why the causal analyst may prefer the use of a proxy model, as mentioned in Appendix C.4: its generative capabilities; its ability to answer multiple causal queries without retraining; and its potential to solve complex identification problems which do not yet have symbolic solutions, such as identification from partially observable distributions. The discussion in Appendix C.4 may also serve as a complement to the discussion above.

---

> > ### Comment · Reviewer_ue4r · 2021-08-27
> > **Two comments on the rebuttal**
> >
> > Thanks for your detailed reply.
> >
> > > it is natural to reason that finite domains can approximate continuous domains in the limit
> >
> > this is however not the setting in which deep neural networks are typically used  in practice, and obviously there is a huge amount of literature to support that when learning from finite samples, continuous domains are challenging in the non-parametric setting. Also classical causal inference results may become tricky to prove in the continuous setting due to measure theoretic issues. Hence I do think it is necessary that you state clearly that your work addresses a very special case with respect to what is currently done in practice, and discuss this limitation in main text. I insist on that point because you did not state explicitly how you will concretely address it in main text. This is not a minor detail but a central aspect for a paper at a top machine learning conference, and we are explicitly asked for checking whether limitations are properly addressed in the paper. Thank you for your understanding.
> >
> > >  many previous works which solve the problems of causal identification and estimation operate on a level of abstraction without the SCM. They typically use the constraints of the causal diagram to symbolically derive a solution (e.g., do-calculus). Instead, this work introduces a framework where one can construct a proxy model to mimic the true SCM (on identifiable quantities) via neural optimization. Thm. 4 and Corols. 2, 4 indicate that identification under our particular proxy model specification, the NCM, is equivalent to symbolic identification through do-calculus. This is a key result in demonstrating that the NCM may be used as a proxy for the true SCM to yield a solution to the problems of causal identification and estimation, the main contribution of our work.
> >
> > I do agree that this should be the key contribution of this work, however, it is not easy to see (in line with the comment of reviewer fTRd, that might reflect the impression of many readers) how far using a proxy makes a difference when such proxy is indistinguishable from the original function on the data and its structure is constrained by the true graph, especially in a context where all quantities are symbolic. In short: in Theorem 4, isn't the identifiability from G-constrained proxies a trivial consequence of identifiability from the original graph in this context? Perhaps providing a simple example where this is not trivial would help.

---

> > > ### Author Response · Authors · 2021-08-29
> > > **Re: Two comments on the rebuttal**
> > >
> > > Thank you for taking the time to provide additional feedback.
> > >
> > > ***
> > > > this is however not the setting in which deep neural networks are typically used in practice, and obviously there is a huge amount of literature to support that when learning from finite samples, continuous domains are challenging in the non-parametric setting. Also classical causal inference results may become tricky to prove in the continuous setting due to measure theoretic issues. Hence I do think it is necessary that you state clearly that your work addresses a very special case with respect to what is currently done in practice, and discuss this limitation in main text. I insist on that point because you did not state explicitly how you will concretely address it in main text. This is not a minor detail but a central aspect for a paper at a top machine learning conference, and we are explicitly asked for checking whether limitations are properly addressed in the paper. Thank you for your understanding.
> > > ***
> > >
> > > **Reply:**
> > > First, as we mentioned before, we appreciate your note that it would be proper to highlight this fact early on. In particular, we plan to add the following text after the definition of SCMs (possibly refined): “In this work, we will assume that the endogenous variables (V) are discrete and have finite domains.” Still, in terms of big picture, we note that the main results in causal inference we are trying to build on and refine are related to identification and estimation, as discussed in the context of Pearl’s do-calculus (Ch. [3, 4] in Causality [Pearl, 2000]), which are mostly concerned with settings when the endogenous variables are discrete, but impose no constraint over the exogenous ones and the structural mechanisms (i.e., neither P(u) nor F of the underlying SCM, respectfully). Having said that, the machinery and results developed here are extendable to the continuous case in natural ways, as alluded to in Footnote 17 (p. 8). So, we will try to add further note on this case based on the discussion here and the current ongoing experiments. All in all, again, we are thankful for your feedback and attention.
> > >
> > > ***
> > > > I do agree that this should be the key contribution of this work, however, it is not easy to see (in line with the comment of reviewer fTRd, that might reflect the impression of many readers) how far using a proxy makes a difference when such proxy is indistinguishable from the original function on the data and its structure is constrained by the true graph, especially in a context where all quantities are symbolic. In short: in Theorem 4, isn't the identifiability from G-constrained proxies a trivial consequence of identifiability from the original graph in this context? Perhaps providing a simple example where this is not trivial would help.
> > > ***
> > >
> > > **Reply:**
> > > This is a good question, which we would be happy to discuss further. Thm. 4 is included out of necessity to establish the NCM as a suitable proxy model for the true SCM M*. Without it (and also Corols. 2 and 4), the identification and estimation results of the NCM would not be meaningful. In practice, this means that one could perform the steps of Alg. 1 on a different model class, but if a similar set of results is not proven for that model class, then he/she cannot conclude that the inferences of that model class will match anything from M*. In other words, your feeling and inclination that identification should be preserved across classes is right, it is just not guaranteed and needs to be proven for the specific model class.
> > >
> > > As perhaps an interesting additional note to this discussion, the NCM model class has many desirable properties (including implying Thm. 4), which are non-trivial and not present in other model classes. For concreteness, we define in Def. 21 (p. 41 in the appendix) another model class called the “L2 expression wrapper”, which is as expressive as the NCM on layer 2. It is essentially a naive example of an L2-representative model class, where we simply fit an independent universal approximator on each L2 distribution. Despite its expressiveness, we show in Example 6 (p. 42) that the L2 expression wrapper is not suitable as a proxy model for M*, since it misses the property of parsimony. Aside from the computational difficulty of training so many independent universal approximators, one must also make sure that all the exponentially many equality constraints implied by G are enforced by the end of training, which is a very hard task for reasonable size models. On the other hand, the inductive bias encoded in the structure of a G-constrained NCM guarantees that these constraints are always maintained without needing special attention in the training process.

---

> > > > ### Comment · Reviewer_ue4r · 2021-09-03
> > > > **Post-discussion feedback**
> > > >
> > > > I would like to thank the reviewers for their useful replies and I am happy with how they plan to revise the manuscript. This improves my perception of the novelty and interest of the work and I am therefore raising my grade to 7.

---

### Decision · Program_Chairs · 2021-09-27

**Decision:**

Accept (Poster)

**Comment:**

The paper considers the Pearl causal hierarchy layers, with $L_1$ representing the observations, $L_2$ the interventions and $L_3$ the counter-factuals.

The contribution establishes that the fact that a powerful model (neural nets in the following (1)) fits the true causal model
${\cal M}^*$ regarding $L_1$, does not imply that it fits ${\cal M}^*$ regarding $L_2$. The proof, following from [6], is based on that fact that the set of models fitting ${\cal M}^*$ on (at least two) layers has measure 0.


In order to tackle the second layer $L_2$ of the Pearl causal hierarchy, one must consider the causal identification task (predicting the causal diagram (2)) and the causal estimation (predicting the effects of intervention).

The second contribution is to establish identifiability conditions (including the knowledge of the causal diagram) for an intervention query.

The algorithmic condition for this identifiability is stated as the fact that the neural models fitting the interventional data and maximizing / minimizing the outcome of the intervention, converge toward each other.

The significance of these contributions has been challenged and very convincingly argued by the authors.

Some limitations regarding the domain of the observed variables (discrete and finite) have been discussed and will be stated in the revised version.

(1) The neural net class is considered in the paper, though the discussion suggests that any class with sufficient expressivity could be considered instead.

(2) The paper claims that there exist no neural methods today focused on solving the causal identification task, which is not exact (though restricted to Markovian), consider https://arxiv.org/abs/1803.04929.